# Photophysical Properties and Kinetic Studies of 2-Vinylpyridine-Based Cycloplatinated(II) Complexes Containing Various Phosphine Ligands [note 1]

**DOI:** 10.3390/molecules26072034

**Published:** 2021-04-02

**Authors:** Vahideh Dolatyari, Hamid R. Shahsavari, Sepideh Habibzadeh, Reza Babadi Aghakhanpour, Sareh Paziresh, Mohsen Golbon Haghighi, Mohammad Reza Halvagar

**Affiliations:** 1Department of Chemistry, Institute for Advanced Studies in Basic Sciences (IASBS), Zanjan 45137-66731, Iran; v.dolatyari@iasbs.ac.ir (V.D.); sareh.paziresh@gmail.com (S.P.); 2Department of Chemistry, Payame Noor University, Tehran P.O. BOX 19395-4697, Iran; spdh.hab@gmail.com; 3Department of Chemistry, Shahid Beheshti University, Evin, Tehran 19839-69411, Iran; m_golbon@sbu.ac.ir; 4Department of Inorganic Chemistry, Chemistry and Chemical Engineering Research Center of Iran, Tehran 14968-13151, Iran; mhalvag@gmail.com

**Keywords:** 2-vinylpyridine, cycloplatinated(II), photophysics, kinetic, phosphine

## Abstract

A series of cycloplatinated(II) complexes with general formula of [PtMe(Vpy)(PR_3_)], Vpy = 2-vinylpyridine and PR_3_ = PPh_3_ (**1a**); PPh_2_Me (**1b**); PPhMe_2_ (**1c**), were synthesized and characterized by means of spectroscopic methods. These cycloplatinated(II) complexes were luminescent at room temperature in the yellow–orange region’s structured bands. The PPhMe_2_ derivative was the strongest emissive among the complexes, and the complex with PPh_3_ was the weakest one. Similar to many luminescent cycloplatinated(II) complexes, the emission was mainly localized on the Vpy cyclometalated ligand as the main chromophoric moiety. The present cycloplatinated(II) complexes were oxidatively reacted with MeI to yield the corresponding cycloplatinated(IV) complexes. The kinetic studies of the reaction point out to an S_N_2 mechanism. The complex with PPhMe_2_ ligand exhibited the fastest oxidative addition reaction due to the most electron-rich Pt(II) center in its structure, whereas the PPh_3_ derivative showed the slowest one. Interestingly, for the PPhMe_2_ analog, the *trans* isomer was stable and could be isolated as both kinetic and thermodynamic product, while the other two underwent *trans* to *cis* isomerization.

## 1. Introduction

Luminescent cyclometalated platinum(II) complexes have been significantly studied by many researchers in the last decades [1,2,3,4,5,6,7,8,9,10,11]; publications in this field of research are increasing [12,13,14,15]. The light-emitting devices [16,17,18] dye-sensitized solar cell [19], photo-switches [20,21], photocatalysts [22], and chemical or biochemical sensors [23] are various applications related to the luminescent cycloplatinated complexes. The cyclometalated ligand is the main chromophoric part in the creation of room-temperature phosphorescence. In addition, heavy metal like Pt induces high spin–orbit coupling and allows singlet-triplet intersystem crossing. In addition to the factors mentioned, the ancillary ligands play a very important role in such complexes’ photophysical properties. These ligands control the electron density at the metal center and consequently the degree of Metal to Ligand Charge Transfer (MLCT) in the lowest energy transition. Some different arrangements for the ancillary ligands are expected depending on the charge (neutral or anionic) or binding mode (chelating and non-chelating) of the ancillary ligands, such as L^X [24,25,26], L^L [6,27,28], L/X [2,29,30,31,32], L/L [33,34] and X/X [35,36].

Kinetic and mechanistic investigations of the oxidative addition reactions of organoplatinum(II) complexes have clearly proved that these reactions follow an S_N_2 mechanism [37,38,39,40,41,42,43,44,45] with the exception of a few cases [46,47]. The S_N_2 mechanism with a large negative ∆S^≠^ value can be observed for the small organic molecules, such as alkyl halides [48,49,50,51]. In these cases, theoretical approaches for mechanistic studies remarkably helped to certify the obtained experimental data [52,53,54,55]. The cycloplatinated(II) complexes with monophosphine ligands are the appropriate complexes for the study of the oxidative addition of alkyl halides on the Pt(II) center [56,57,58,59,60,61,62]. The phosphine ligands can affect electronically and sterically the rate constants and activation parameters. Electron-withdrawing or -donating properties of the substituents together with the Tolman cone angle significantly control the properties of monophosphine ligand [56]. In a comparison between PPh_2_Me and PPh_3_ ligands, the rate constants attributed to the PPh_2_Me derivative are 3‒5-fold more than those for PPh_3_, indicating the electronic and steric differences [63]. In this manner, it is expected that the PPhMe_2_ derivative reactions should be considerably faster than those of PPh_2_Me.

In the framework of our experiences on the photophysical [30,31,64,65,66,67,68,69] and the kinetico-mechanistic studies [70,71,72,73,74] of cyclometalated platinum complexes, some cycloplatinated(II or IV) complexes with 2-vinylpyridine (Vpy) and phosphine ligands (PPh_3_ [75], PPh_2_Me [76] and PPhMe_2_) were designed. The kinetic and photophysical properties of the complex bearing PPh_2_Me were previously reported by us [76]. Therefore, herein, we added the PPh_3_ and PPhMe_2_ derivatives to make a meaningful trend and have a complete picture for comparison in terms of photophysical and kinetic studies. In addition, density functional theory (DFT) calculations were performed for both investigations and support the experimental data.

## 2. Result and Discussion

### 2.1. Synthesis and Characterization

All the synthetic steps are demonstrated in Scheme 1. In this scheme, the precursor cycloplatinated(II) complex [PtMe(Vpy)(DMSO)], **A**, [75,76,77] Vpy = 2-vinylpyridine, reacted with several monophosphines to give the complexes [PtMe(Vpy)(PR_3_)], PR_3_ = PPh_3_ (**1a**) [75]; PPh_2_Me (**1b**) [76]; PPhMe_2_ (**1c**). These complexes treated with MeI to give the corresponding cycloplatinated(IV) complexes of *trans*-[PtMe_2_I(Vpy)(PR_3_)], PR_3_ = PPh_3_ (**2a**); PPh_2_Me (**2b**) [76] PPhMe_2_ (**2c**), and then *cis*-[PtMe_2_I(Vpy)(PR_3_)], PR_3_ = PPh_3_ (**3a**); PPh_2_Me (**3b**) [76]; PPhMe_2_ (**3c**). We have previously reported **1b** and its reaction with MeI (to form **2b** and **3b**) kinetically investigated [76]. The Pt(IV) complex **2b** was not stable and gradually converted to **3b** isomer, and it was decomposed to the known Pt(II) complex and an organic compound [76]. The same interconversion happened for **1a** with this difference that **2a** (*trans* isomer) cannot be detected due to the very fast *trans* to *cis* isomerization. The *cis* isomer **3a** is similarly decomposed and forms the same products (Scheme 2; *trans*-[PtMeI(PPh_3_)_2_], **4**, [{PtMe_2_(Vpy)}_2_(*µ*-I)_2_], **5**, and *Z*/*E*-[C_8_H_9_N], **6**) [76]. On the other hand, for **1c,** the corresponding cycloplatinated(IV) **2c** is stable and not converted to **3c,** and consequently, no decomposition process occurs (Scheme 2). The above-mentioned observations decisively point out the determining role of the phosphine ligand in determining the stability of the complexes.

The formation of the new complexes **1c**, **2c** and **3a** confirmed by ^1^H- and ^31^P{^1^H}-NMR spectroscopy and their spectra have all been shown in Appendix A. The structures of **1c** and **2c** were further characterized by X-ray crystallography (Appendix A), and their Oak Ridge Thermal Ellipsoid Plot (ORTEP) plots are shown in Figure 1, while their selected geometrical parameters are presented in its caption. The structure of **1c** clearly confirms the proposed structure, and as expected, the C ligating atom of Vpy is located *trans* to the P of PPhMe_2_. The Vpy bite angle in **1c** (N1-Pt1-C11) is 78.76°, which is very close to those observed for the other cyclometalated ligands in Pt(II) complexes [6]. Upon oxidative addition reaction, this angle in **2c** does not show a meaningful change (79.09°). It can vividly be observed that the *trans* isomer is stable for **2c** [72].

### 2.2. Photophysical Properties

The electronic transitions of the cycloplatinated(II) complexes **1a**‒**c** were initially investigated by the UV-vis spectroscopy, and their spectra were characterized in detail by the help of density functional theory (DFT) and time dependence-DFT (TD–DFT) calculations. In the first step, the ground states of all these three complexes were optimized with the consideration of CH_2_Cl_2_ as the solvent (the same solvent for experimental absorption spectra); the optimized structures are shown in Appendix A, and the selected geometrical parameters are listed in Appendix A. Using the optimized structures in CH_2_Cl_2_, the frontier molecular orbitals (MO), including “HOMO to HOMO−5” (Highest Occupied Molecular Orbital) and “LUMO to LUMO+5,” (Lowest Unoccupied Molecular Orbital) were obtained for all the complexes. The compositions of the selected molecular orbitals in terms of Pt and ligands are listed in Appendix A, and the corresponding visual plots are depicted in Appendix A. For all the cases, HOMO is predominantly localized on Pt and Vpy fragments, while in LUMO, Vpy is significantly dominated over the other fragments. In addition, the contribution of phosphine ligand is considerably increased in lower HOMOs and higher LUMOs.

As observed in Figure 2, although there is an acceptable agreement between the experimental UV-vis spectra and corresponding TD–DFT bars, an overestimation for the transition energies is observed in the visible region for all the cases. Table 1 indicates that the low-energy bands (the wavelengths used for yielding the emission bands) in all the complexes are attributed to the electronic transitions in the cyclometalated ligand and also charge transfer from Pt to the cyclometalated ligand (mixed ^1^ILCT/^1^MLCT (Intra Ligand Charge Transfer), L = Vpy). In other words, the first excited state (S1) is majorly related to the HOMO **→** LUMO transitions, which are assigned as ^1^MLCT/^1^ILCT. However, in higher energy bands (higher excited states) and the ^1^ILCT and ^1^MLCT, ^1^ML′CT (L′ = PR_3_) and ^1^LL′CT (Ligand to Ligand Charge Transfer) electronic transitions are present, which involve the transitions with the phosphine ligand as the target.

The photophysical properties of **1a**–**c** were examined using photoluminescence spectroscopy. All three complexes are luminescent in solid state at room temperature and low-temperature, having the bands in yellow-orange area (see Table 2 for the numerical emission parameters and Figure 3 for the emission spectra). Expectedly, the complexes exhibit brighter emissions at 77 K in relation to 298 K, which is due to the more rigidity of the structures at low-temperature (see Table 2 for the quantum yield (QY) values). The lifetime values measured for the complexes at both temperature conditions are in microsecond scale, indicating phosphorescence character in the emissive states. The absolute QY values of the complexes obey the trend **1c** > **1b** > **1a**. This is probably related to the more electron-donating character of PPhMe_2_ compared to PPh_2_Me and PPh_3_, which makes the Pt(II) center and consequently the chromophoric ligand of Vpy to be more electron-rich. As can be seen in Figure 3, the emission spectra for the entire cases exhibit structured emission bands, which always point out that the emission is majorly localized on the cyclometalated ligands (large amount of ^3^ILCT in the emissive state together with the small amount of ^3^MLCT) [6,30]. The mixed ^3^ILCT/^3^MLCT character for the emission bands reflects the ^1^ILCT/^1^MLCT character the in absorption spectra obtained by the TD–DFT calculations. The wavelength of emissions, being 550 nm at 298 K for all the complexes, is another evidence to confirm the identical emission nature in all the complexes (Vpy ligand). Upon lowering the temperature, a marginal blue shift is observed for the low-temperature bands in relation to their parents at room temperature, and also no tangible change is observed in the color or character of emissions. It should be noted that this blue shift is slightly larger for **1a** compared to the other derivatives.

### 2.3. Kinetic Studies

The oxidative addition of an excess amount of MeI on **1a** and **1c** in acetone solvent was kinetically studied by following the disappearance of the ^1^MLCT band of the cycloplatinated(II) complexes in the absorption spectra (see Figure 4 for the changes in the UV-vis spectra of **1a** and **1c** at room temperature). The parameter *k*_obs_ (pseudo-first-order rate constant) was calculated by the nonlinear least-squares fitting of the absorbance-time curves to a first-order equation. The *k*_obs_/MeI concentration graphs were plotted for at least five temperatures (Figure 4), giving desirable straight-line plots with no intercept (no involvement of solvent or any dissociative pathway). The slope of the straight lines gives the second-order rate constant (*k*_2_). To calculate the activation parameters, the Eyring equation (Equation (1)) was employed at different temperatures (Figure 5). The large negative Entropy of Activation Δ*S*^#^ values in Table 3 are normally indicative of second-order kinetics (first-order for Pt(II) complex and MeI) and S_N_2 mechanism. We have previously suggested an S_N_2 mechanism for **1b,** and the present reactions for **1a** and **1c** obey the previously reported mechanism.^76^ In this mechanism, the *trans* isomer is considered as the kinetic product, while the thermodynamic product is attributed to the *cis* isomer. At all the temperatures, the reaction of **1c** (containing PPhMe_2_) with MeI is the fastest reaction, and **1b** and **1a** are the lower ranks, respectively. Therefore, the Enthalpy of Activation Δ*H*^#^ value for the reaction of **1c** with MeI is the lowest value (the lowest energy barrier). However, the Δ*S*^#^ values are controlled by the steric hindrance induced by phosphine ligands. The highest Δ*S*^#^ value is attributed to the **1c** having PPhMe_2_ with the lowest steric hindrance. This is due to the associative mechanism with a penta-coordinated intermediate in which the lowest steric hindrance makes the highest Δ*S*^#^ value.

From Figure 5 and Table 3, it can be understood that the rate of the reactions obeys the trend **1c** (PPhMe_2_) > **1b** (PPh_2_Me) > **1a** (PPh_3_). For example, at 25 °C, the rate for **1b** is almost 5 times larger than that for **1a,** while the reaction of **1c** is 7 times faster than that of **1b**. This is absolutely related to the electron-donating trend of the present monophosphine ligands (PPhMe_2_ > PPh_2_Me > PPh_3_). The more electron-donating ability makes more electron-rich Pt(II) center, which is favorable for oxidative addition reaction and makes it to be faster. Therefore, in the formation of kinetic products, the electronic factors are effective. However, in trans to *cis* isomerization (kinetic product to thermodynamic product), the steric effects play a determining role. Additionally, the *trans* isomer can be stabilized by the electron-donating phosphine ligands like PPhMe_2_ [78] For the reaction of **1a** with MeI, the kinetic product (**2a**) cannot be observed, and only the *cis* isomer (**3a**) as the thermodynamic product can be isolated. This *trans* to *cis* isomerization is attributed to a large steric hindrance and low electron-donating ability of the PPh_3_ ligand. In the case of the reaction of **1b** with MeI [76] having a phosphine (PPh_2_Me) with less steric hindrance and more electron-donating ability compared with PPh_3_, the *trans* isomer can be isolated and even completely characterized, but it will gradually be converted to the *cis* product and eventually decomposed [74]. However, if PPh_2_Me is replaced by PPhMe_2_, due to the smaller steric hindrance and large electron-donating ability imposed by PPhMe_2_, the *trans* to *cis* isomerization is practically prohibited and *trans* isomer **2c** as the kinetic product is completely stable so that it can be crystallized and characterized by X-ray crystallography.

## 3. Experimental

### 3.1. General Procedures and Materials

^1^H-NMR (400 MHz) and ^31^P{^1^H}-NMR (162 MHz) spectra were recorded on a Bruker Avance III instrument (Ettlingen, Germany) and are referenced to the external standards, i.e., SiMe_4_ and 85% H_3_PO_4_, respectively. The chemical shifts (δ) being reported as ppm and coupling constants (*J*) expressed in Hz. UV-vis absorption spectra and kinetic studies were carried out by using an Ultrospec 4000 Pro, UV-vis spectrometer (Little Chalfont, UK) with temperature control using a Pharmacia Biotech constant-temperature bath. Excitation and emission spectra were obtained on a PerkinElmer LS45 fluorescence spectrometer (Beaconsfield, UK) with the lifetimes measured in phosphorimeter mode, and the quantum yields of the complexes were measured using an integrating sphere. 2-Vinylpyridine (Vpy), triphenylphosphine (PPh_3_), dimethylphenylphosphine (PPhMe_2_) and the other chemicals were purchased from Sigma Aldrich (St. Louis, MO, USA). All of the reactions were carried out under an argon atmosphere, and all of the common solvents were purified and dried according to standard procedures. The complexes [PtMe(Vpy)(DMSO)], **A**, [PtMe(Vpy)(PPh_3_)], **1a**, [PtMe(Vpy)(PPh_2_Me)], **1b**, and [PtMe_2_I(Vpy)(PPh_2_Me)], **2b**, were prepared according to literature methods [75,76,77]. The NMR labeling for the Vpy ligand for clarifying the chemical shift assignments is shown in Scheme 3. New NMR data for **1a**, ^1^H-NMR (400 MHz, acetone-*d*_6_, 295 K): δ 0.69 (d, ^3^*J*_PH_ = 8.1, ^2^*J*_PtH_ = 84.2 Hz, 3H, PtMe), 6.57 (t, ^3^*J*_HH_ = 6.3 Hz, 1H, H^5^), 7.25 (d, ^3^*J*_HH_ = 7.8 Hz, 1H, H^3^), 7.33 (dd, ^3^*J*_HH_ = 9.2 Hz, ^4^*J*_PH_ = 15.2 Hz, ^3^*J*_PtH_ = 94.3 Hz, 1H, H*^α^*), 7.44–7.49 (m, 9H, H*^m^* and H*^p^* PPh_3_), 7.58 (d, ^3^*J*_HH_ = 5.3 Hz, ^3^*J*_PtH_ = 16.4 Hz, 1H, H^6^), 7.68–7.74 (m, 7H, H^4^ and H*^o^* PPh_3_), 7.76 (dd, ^3^*J*_HH_ = 9.2 Hz, ^3^*J*_PH_ = 7.8 Hz, ^2^*J*_PtH_ = 163.4 Hz, 1H, H*^β^*); ^31^P{^1^H}-NMR (162 MHz, acetone-*d*_6_, 295 K): δ 31.8 (s, ^1^*J*_PtP_ = 2021 Hz, 1P).

### 3.2. Synthesis of Complexes

#### 3.2.1. [PtMe(Vpy)(PPhMe_2_)], **1c**

To a solution of [PtMe(Vpy)(DMSO)], **A**, (150 mg, 0.382 mmol) in acetone (15 mL) was added PPhMe_2_ ligand (60 μL, 0.420 mmol, 1.1 equivalent). The mixture was stirred at room temperature for 2 h. The deep orange solution was concentrated to a small volume (~2 mL), and *n*-hexane was added (3 mL) to give an orange solid identified as **1c**. Yield: 93%; Anal. Calcd for C_16_H_20_NPPt (452.39); C, 42.48; H, 4.46; N, 3.10. Found: C, 42.91; H, 4.56; N, 3.17. ^1^H-NMR (400 MHz, acetone-*d*_6_, 295 K): δ 0.95 (d, ^3^*J*_PH_ = 8.7, ^2^*J*_PtH_ = 85.3 Hz, 3H, PtMe), 1.71 (d, ^2^*J*_PH_ = 7.7, ^3^*J*_PtH_ = 20.9 Hz, 6H, Me of PPhMe_2_), 6.77 (td, ^3^*J*_HH_ = 6.5 Hz, ^4^*J*_HH_ = 1.4 Hz, 1H, H^5^), 7.20 (d, ^3^*J*_HH_ = 7.9 Hz, 1H, H^3^), 7.24 (dd, ^3^*J*_HH_ = 9.4 Hz, ^4^*J*_PH_ = 15.7 Hz, ^3^*J*_PtH_ = 87.1 Hz, 1H, H*^α^*), 7.45–7.48 (m, 3H, H^4^ and H*^m^* PPhMe_2_), 7.72 (t, ^3^*J*_HH_ = 8.2 Hz, 1H, H*^p^* PPhMe_2_), 7.73 (t, ^3^*J*_HH_ + ^3^*J*_PH_ = 9.1 Hz, 7.8 Hz, ^2^*J*_PtH_ = 149.1 Hz, 1H, H*^β^*), 7.89–7.95 (m, 3H, H^6^ and H*^o^* PPh_3_); ^31^P{^1^H}-NMR (162 MHz, acetone-*d*_6_, 295 K): δ −4.6 (s, ^1^*J*_PtP_ = 1967 Hz, 1P).

#### 3.2.2. *cis*-[PtMe_2_I(Vpy)(PPh_3_)], **3a**

To solution of [PtMe(Vpy)(PPh_3_)], **1a**, (100 mg, 0.173 mmol) in acetone (15 mL) were added 250 μL (excess, 25-fold) of MeI. The solution was stirred for 3 h at room temperature, then diethyl ether was added to give a precipitate, which was filtered, washed with diethyl ether to give the product as a pale yellow solid identified as **3a**. The product was dried in vacuum. Yield: 79%; Anal. Calcd for C_27_H_27_INPPt (718.47); C, 45.14; H, 3.79; N, 1.95. Found: C, 45.32; H, 3.87; N, 1.82. ^1^H-NMR (400 MHz, acetone-*d*_6_, 295 K): 1.04 (d, ^3^*J*_PH_ = 7.8, ^2^*J*_PtH_ = 60.7 Hz, 3H, Me ligand *trans* to PPh_3_, PtMe), 1.75 (d, ^3^*J*_PH_ = 7.9, ^2^*J*_PtH_ = 69.8 Hz, 3H, Me ligand *trans* to N of Vpy, PtMe), 6.65 (dd, ^3^*J*_HH_ = 6.9 Hz, ^4^*J*_PH_ = 2.6 Hz, ^3^*J*_PtH_ = 104.2 Hz, 1H, H*^α^*), 6.95 (t, ^3^*J*_HH_ = 6.7 Hz, 1H, H^5^), 7.31–7.45 (m, 9H, H*^m^* and H*^p^* PPh_3_), 7.51–7.96 (m, 9H, H*^β^*, H^3^, H^4^ and H*^o^* PPh_3_), 9.07 (d, ^3^*J*_HH_ = 5.1 Hz, ^3^*J*_PtH_ = 13.3 Hz, 1H, H^6^); ^31^P{^1^H}-NMR (162 MHz, acetone-*d*_6_, 295 K): δ −7.8 (s, ^1^*J*_PtP_ = 1036 Hz, 1P).

#### 3.2.3. *trans*-[PtMe_2_I(Vpy)(PPhMe_2_)], **2c**

To solution of [PtMe(Vpy)(PPhMe_2_)], **1c**, (100 mg, 0.221 mmol) in acetone (15 mL) were added 150 μL (excess, 10-fold) of MeI. The solution was stirred for 3 h at room temperature, then diethyl ether was added to give a precipitate, which was filtered, washed with diethyl ether to give the product as a pale yellow solid identified as **2c**. The product was dried in vacuum. Yield: 81%; Anal. Calcd for C_17_H_23_INPPt (594.33); C, 34.36; H, 3.90; N, 2.36. Found: C, 34.25; H, 3.92; N, 2.33. ^1^H-NMR (400 MHz, acetone-*d*_6_, 295 K): 0.65 (d, ^3^*J*_PH_ = 7.6, ^2^*J*_PtH_ = 70.3 Hz, 3H, Me ligand *trans* to I, PtMe), 1.26 (d, ^3^*J*_PH_ = 8.2, ^2^*J*_PtH_ = 69.4 Hz, 3H, Me ligand *trans* to N of Vpy, PtMe), 1.98 (d, ^2^*J*_PH_ = 8.9, ^3^*J*_PtH_ = 12.6 Hz, 3H, Me of PPhMe_2_), 2.21 (d, ^2^*J*_PH_ = 9.0, ^3^*J*_PtH_ = 11.1 Hz, 3H, Me of PPhMe_2_), 6.83 (td, ^3^*J*_HH_ = 6.6 Hz, ^4^*J*_HH_ = 1.3 Hz, 1H, H^5^), 7.03 (dd, ^3^*J*_HH_ = 8.0 Hz, ^4^*J*_PH_ = 21.7 Hz, ^3^*J*_PtH_ = 66.9 Hz, 1H, H*^α^*), 7.44 (d, ^3^*J*_HH_ = 7.8 Hz, 1H, H^3^), 7.52–7.59 (m, 3H, H*^p^* and H*^m^* PPhMe_2_), 7.72 (td, ^3^*J*_HH_ = 7.7 Hz, ^4^*J*_HH_ = 1.5 Hz, 1H, H^4^), 7.81 (d, ^3^*J*_HH_ = 5.1 Hz, ^3^*J*_PtH_ = 12.0 Hz, 1H, H^6^), 7.86–7.92 (m, 2H, H*^o^* PPh_3_), 8.03 (dd, ^3^*J*_HH_ = 8.0 Hz, ^3^*J*_PH_ = 9.2 Hz, ^2^*J*_PtH_ = 75.6 Hz, 1H, H*^β^*); ^31^P{^1^H}-NMR (162 MHz, acetone-*d*_6_, 295 K): δ −37.8 (s, ^1^*J*_PtP_ = 1278 Hz, 1P).

### 3.3. Kinetic Study

A solution of **1a** or **1c** in acetone (3 mL, 2.5 × 10^−4^ M) in a cuvette was thermostated at 25 °C, and a known excess of MeI (150 μL, 3200-fold for **1a**; 15 μL, 320-fold for **1c**) was added using a microsyringe. After rapid stirring, the absorbance at λ = 387 (**1a**) or 386 (**1c**) nm was collected with time. The absorbance-time curves were analyzed by the pseudo-first-order method (A_t_ = (A_0_ − A_∞_) exp(−*k*_obs_t) + A_∞_). The same method was used at other concentrations, and temperatures (10, 15, 25, and 30 °C) and activation parameters (ΔS^#^ and ΔH^#^) were obtained from the Eyring equation (Equation (1)) and the full data are collected in Table 1. Ref. [79]
(1)$$\ln\left(\frac{k_{2}}{T}\right)\;=\;\ln\left(\frac{k_{B}}{h}\right)\;+\;\frac{\Delta S^{‡}}{R}\;-\;\frac{\Delta H^{‡}}{\mathrm{RT}}.$$

### 3.4. X-ray Crystallography

The X-ray diffraction measurement was carried out on a STOE IPDS2T diffractometer (STOE & Cie GmbH, Darmstadt, Germany) with graphite-monochromated Mo K*α* radiation. The single crystals suitable for X-ray analysis were obtained from CH_2_Cl_2_/*n*-hexane solution (at room temperature) and mounted on glass fiber, and used for data collection. Cell constants and an orientation matrix for data collection were obtained by least-square refinement of the diffraction data for **1c** and **2c**. Diffraction data were collected in a series of ω scans in 1° oscillations and integrated using the Stoe X-AREA software package (Stoe & Cie GmbH, Darmstadt, Germany) [80]. Numerical absorption correction was applied using X-Red32 software (Stoe & Cie GmbH, Darmstadt, Germany). The structure was solved by direct methods and subsequent difference Fourier maps and then refined on F2 by a full-matrix least-squares procedure using anisotropic displacement parameters. Atomic factors are from the International Tables for X-ray Crystallography. All non-hydrogen atoms were refined with anisotropic displacement parameters. Hydrogen atoms were placed in ideal positions and refined as riding atoms with relative isotropic displacement parameters. All refinements were performed using the X-STEP32, SHELXL-2014 and WinGX-2013.3 programs [81,82,83,84,85] Crystallographic data for the structural analysis was deposited with the Cambridge Crystallographic Data Centre, No. CCDC-2057874 (for **1c**) and CCDC-2057873 (for **2c**).

### 3.5. Computational Details

Density functional calculations were performed with the program suite Gaussian09 (Wallingford, CT, USA) [86] using the B3LYP level of theory [87,88,89]. The LANL2DZ basis set was chosen to describe Pt [90,91] and the 6-31G(d) basis set was chosen for other atoms. The geometries of complexes were fully optimized by employing the density functional theory without imposing any symmetry constraints. To ensure the optimized geometries, frequency calculations were performed employing analytical second derivatives. CH_2_Cl_2_ as solvent was introduced using the conductor-like polarizable continuum model (CPCM) [92,93]. The calculations for the electronic absorption spectra by time-dependent DFT (TD–DFT) were performed at the same level of theory. The compositions of molecular orbitals and theoretical absorption spectra were plotted using the “Chemissian” software (Petersburg, Russia) [94].

## 4. Conclusion

We have reported synthesis, photophysical properties of **1b**, and kinetic investigation of its reaction with MeI [76]. In this work, we completed the trend of phosphine ligands by synthesis of **1a** and **1c**. The effect of phosphines on the photophysical properties of the cycloplatinated(II) complexes and also on the rates of oxidative addition reactions with MeI studied. These complexes show emission bands in the yellow-orange region with the same wavelength of 550 nm. The emission bands exhibit a structured shape, which is indicative of the mixed character ^3^ILCT/^3^MLCT and emission from the Vpy cyclometalated ligand. However, **1c** shows the strongest emission, while **1a** shows the weakest emission, which is probably related to the electron-donating ability of the phosphine ligands, which obeys the trend PPhMe_2_ > PPh_2_Me > PPh_3_. On the other hand, the nature of phosphine ligands affects the rate of oxidative addition reactions of cycloplatinated(II) complexes with MeI. The expected rate trend of **1c** > **1b** > **1a** was obtained, which is attributed to the electron-donating ability of the phosphine ligands (PPhMe_2_ > PPh_2_Me > PPh_3_), being favorable for the oxidative addition reaction. In this regard, *trans* to *cis* isomerization is also observed for two cases of **2a** (very fast) and **2b** (slow), which is controlled by the steric hindrance induced by the phosphine. However, **2c** as the *trans* isomer and kinetic product is stable and can be isolated and crystallized, which is due to the small steric hindrance of the PPhMe_2_ ligand.

## Data Availability

Data is contained within the article or Appendix A.

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
