# Peer review of "Photophysical Properties and Kinetic Studies of 2-Vinylpyridine-Based Cycloplatinated(II) Complexes Containing Various Phosphine Ligands†"

_molecules, 2021, doi:10.3390/molecules26072034_

Round 1

Reviewer 1 Report

Comments to authors on manuscript: molecules-1142257

The manuscript by Dolatyari and co-workers reports kinetic data for the oxidative addition of methyl iodide to a series of isostructural k2-N,C-vinylpyridine methyl complexes of square-planar platinum(II) that differ from each other by a tertiary phosphine ligand. Successive substitution of phenyl for methyl groups along the series PPh3-nMen (n=0,1,2) renders the platinum(II) complex more reactive toward the oxidative addition of MeI. The type of phosphine not only affects the rate constant for oxidative addition of MeI but also has an impact on the conversion of initially formed six-coordinate Pt(IV) product complexes into thermodynamically favored isomeric structures as well as their degradation into a mixture of organometallic and organic products. Within the series of platinum(II) precursors, increasing sigma-donor capability among the series of phosphines associates with increasing lifetimes of electronically excited states and photoluminescence quantum yields.

In terms of scientific quality, this work fulfills the requirements for publication in Molecules but a minor revision is strongly recommended prior to final acceptance. The following points should be addressed:

  1. Page 4 of 16 reports TD-DFT derived electronic transitions to be in ‘acceptable agreement’ with experimental data. Inspection of Figure 2a-2c leads to the impression of a rather poor match for the visible spectral region, which however is the spectral region that is relevant to the interpretation of photoluminescence properties. Authors should provide justification for their assessment.
  2. Authors provide activation parameters for the oxidative addition reaction but there is no discussion of the distinct opposite trend of activation enthalpy and entropy. In addition, errors provided for DH# and DS# are much too small, in particular that for DS# considering that the data rely on the temperature range of only 20°C.
  3. Page 6 of 16, line 173: A largely negative activation entropy is not generally ‘indicative of a second order kinetics’.
  4. The take-home message of Figure 5 is unclear and authors should add reference to their previous work.
  5. Discussion of the stability of the initial reaction products from MeI oxidative addition lacks experimental support. Authors are encouraged to make use of experimental activation parameters for this purpose. The opposite trend of DH# and DS# may be considered to reflect the different steric demands of phosphine ligands. In preparation of the transition state for the bimolecular reaction with MeI, a greater steric demand of the phosphine appears to associate with greater activation enthalpy for elongating the Pt-P bond, which connects to a more favorable activation entropy and vice versa.
  6. There is a sudden change of focus in the Introduction at the transition from page 1 to 2, namely from photoluminescence properties to oxidative addition reactivity of platinum(II) compounds. The relation of these different topics is not immediately clear to readers.
  7. Abstract and main text: ‘center of emission’ should be revised
  8. Page 1 of 16: ‘amount of MLCT’ is not scientifically sound and should be revised
  9. Page 2 of 16, lines 77-78: Including a reference to Scheme 2 here helps the reader to get the meaning
  10. Scheme 1: In fact, all complexes are ‘cis’ isomers. Reference to cis-trans isomerization (typo in Scheme 1: ‘isomeration’) does not fit Scheme 1.
  11. Numbers of products that were not observed or substantiated should be removed from Scheme 1 to avoid misunderstandings. A brief statement added to scheme title may clarify the point.
  12. Page 2 of 16, line 85: ‘… in determination of the stable geometries of the complexes’. What is meant here, kinetic or thermodynamic stability?
  13. Page 3 of 16, line 111: ‘optimized in CH2Cl2’ should be revised, giving proper information as that provided in the computational details
  14. Figure 4: Please improve font sizes and colours because information is hardly legible even on magnification on the screen. In addition, pasting a downward arrow into the series of spectra will aid the reader to perceive data more easily.

Author Response

Dear Reviewer,

We would like to thank you very much for giving us the opportunity to revise our above submission. We also greatly appreciate the constructive comments by the respectable referees. The comments raised by them have been carefully addressed and the manuscript has been accordingly revised and highlighted in BLUE color in the revised manuscript. The answers to the editorial office and the referees’ comments are as described below. We hope that you find this revised manuscript suitable for publication in Molecules. We would appreciate your consideration of the matter.

Sincerely,

Hamid R. Shahsavari

Reviewer 2 Report

This manuscript reports a series of three cyclometalated vinylpyridine complexes of Pt(II) featuring three different phosphine ancillary ligands. The complexes are emissive in the solid-state, with photophysical properties being reported and supported by complimentary computational studies. The oxidative addition of MeI to the complexes is investigated, leading to the observation that the more electron donating phosphine ligands lead to a faster rate of addition. Complexes 1a and 1b have been reported previously (Eur. J. Inorg. Chem., 2014, 2278; J. Organomet. Chem. 2016, 803, 82) which somewhat reduces the novelty of this manuscript; however, the inclusion of 1c and a thorough photophysical and kinetic investigation of all complexes (previously unreported) merits the publication of this work in Molecules.

I would recommend acceptance of this manuscript for publication in Molecules subject to the authors addressing the following, fairly minor, points:

  1. Line 35.  The phrase 'most important factor' is not clear. This should be re-phrased in the context of the specific electronic properties of the Vpy ligand.
  2. Lines 94/95.  The comparison of Vpy bite-angle to other Pt(II) complexes requires a literature citation.
  3. UV-Vis absorbance profiles should be presented showing molar extinction coefficient values on the y-axis rather than arbitrary units.
  4. I would not entirely agree that there is an 'acceptable agreement' between the TD-DFT calculated transitions and the experimental spectra. Whilst I do not doubt the qualitative validity of the calculations, there is a clear over-estimation of transition energies. This should be duly noted within the text.
  5. All emission spectra are reported for solid samples, which without justification is somewhat unusual, especially given that the authors have previously reported the luminescence of 1b in CH2Cl2 fluid solution. To compliment this earlier report, the authors should also report the luminescence spectra, lifetimes and PLQY values for 1a-3a in fluid CH2Cl2 solution. Even if this is confined to the supporting information, if would make a worthwhile and valuable addition to the current manuscript.
  6. Table 2. There is a typing mistake on the bottom row.
  7. Figure 3a. The pronounced peak at 450 nm in the excitation spectrum recorded at 77K is unexpected. This is not present for the other complexes. The authors should comment upon this. It is possible that this indicates the presence of an impurity in the sample. Indeed, an additional resonance in the corresponding 31P NMR shown in Fig. S5 suggests that another phosphorus-containing species is present.
  8. The kinetic experiments are thorough and well presented. Perhaps the use of a 10-fold difference in excess MeI between 1a (3200 eq.) and 1c (320 eq.) could be justified.
  9. Line 268. 2c should be 3a.
  10. Line 290. The 31P chemical shift should be -37.8 ppm, according to Fig. S8

Author Response

(The authors gave the same response as above.)

Reviewer 3 Report

In this work Dolatyari at alii reported a further piece of their extensively work on the chemistry of the cyclometaled Pt(II) complexes. In particular, the work is closely related to the ones of reference 82 and 83 applying the same methodology of investigation on a third compounds of increased electron donating characteristic (PPhMe2 > PPh2Me > PPh3)

As stated in the introduction, the goal was to highlight possible trends and to better understand the effects of the phosphine ancillary ligands on the photophisical property and on the reactivity of 2-vinylpyridine Pt(II) complexes.

A correlation between the electron-donating ability of the phosphines and both the emission performances and the reaction rate of the oxidative addiction of MeI is reported.

Finally, the authors also reported the existence of two isomers. A trans complex is the direct result of the SN2 reaction with MeI (the kinetic form), which may interconvert to the thermodynamic cis form.

This latter subsequently undergoes decomposition. The authors found that the isomerization process is ruled by the nature of the phosphine ligands, in particular, in the case of PPhMe2 the interconvertion to the cis form was not observed thus preventing the decomposition of the complex.

The paper reports interesting results, however critical aspects need to be addressed before publication.

As general comment the English language is poor in several parts of the text and needs revision. In some sentences the linguistic register is too low and should be revised (eg. page 2 line 78-79 The same story happened…)

A second concern is about the number of self-citations, which are more than 1/3 of the total references. I’m aware of the extensive production of the authors in the field but in some points more relevant citations were excluded in favor of citations by the authors. I’m not suggesting removing the authors’ work but to acknowledge the contributions of other groups to the field. For instance, in page 2 line 45 three of the four citations used for the oxidative addition reactions are from the authors. Surprisingly the Organometallics, 1999, 18, 2116 cited by the authors in other papers was left out.

Turning to more specific issues:

- page 1 line 34, in my opinion the sentence “The cyclometalated ligand is the most important factor in creation of room temperature phosphorescence which needs a heavy metal like Pt to induce high spin-orbit coupling and allow singlet-triplet intersystem crossingprocess.” needs to be revised.

- page 2 line 96, is the stability depending on the bite-angle? As it is written, the sentence is not clear.

- did the authors attempt the optimization of T1 either by TD-DFT or unrestricted calculations? The absorption was characterized by the mean of DFT/TD-DFT calculations, but only a brief comment was used to describe the emitting state. Either it should be better express the relation between figure 3 and the nature of the emission transition or some citations are required.

- The authors claim that the trans/cis isomerization is ruled by the steric effect of the phosphine substituent. Other authors suggested that in octahedral Ru(II) complexes the isomerization from a trans to a cis form is guided by the basicity of the diphosphine ligands (Facchetti, G., Cesarotti, E., Pellizzoni, M., Zerla, D. and Rimoldi, I. (2012), Eur. J. Inorg. Chem., 2012: 4365-4370.), more electron rich phosphines favored the trans isomer.

This could also be the case in the present study, where the more electron rich phosphine (PPhMe2) leads to the observation of only the trans form. Why are the author so sure that only the steric hindrance play a role in the isomerization process? Can the author clarify this point?

A last curiosity, did the authors try to push the isomerization of 2c toward the cis form for instance by increasing the reaction temperature or changing the solvent?

Author Response

(The authors gave the same response as above.)

Reviewer 4 Report

The manuscript entitled "Photophysical Properties and Kinetic Studies of 2-Vinylpyri-2 dine-Based Cycloplatinated(II) Complexes Containing Various 3 Phosphine Ligands" studies new cyclometalated platinum(II) complexes by means of spectroscopic and computational methods. The manuscript it quite well written, though requires some language corrections. In general the study is performed quite well and the conclusions are well supported by the results. The problem with this study is, however, that it completely unoriginal and shows only three new complexes. Some older studies dealing with very similar complexes (e.g. 10.1016/j.jorganchem.2016.05.005, cited in the text) also present a limited number of new complexes, but are much more detailed in the computational part, suggesting a mechanism for oxidation of Pt(II) to Pt(IV) complexes. This study is basically a copy of that older investigation with a slight modification of the complexes in question - such a slight one, that one could in principle deduct the results without performing any experiments, as these are also very similar to the previously obtained ones.

As such I don't see a point in publishing such a similar study with so little novelty in Molecules and I would advise either a) synthesizing and characterizing a lot more complexes so at least some statistics or structure-activity relationships studies can be performed or b) changing a journal to a much lower-impact one that focuses on structural chemistry.

Author Response

(The authors gave the same response as above.)

Round 2

Reviewer 4 Report

The improved manuscript entitled "Photophysical Properties and Kinetic Studies of 2-Vinylpyri-2 dine-Based Cycloplatinated(II) Complexes Containing Various 3 Phosphine Ligands" adds a few very similar complexes to the original study. While its a nice addition, it does not really fix the main problem of this investigation, that it completely unoriginal. This study is basically a copy of that older investigation (cited in the text) with a slight modification of the complexes in question - such a slight one, that one could in principle deduct the results without performing any experiments, as these are also very similar to the previously obtained ones.

As such I still don't see a point in publishing such a similar study with so little novelty in Molecules and I would advise changing a journal to a much lower-impact one that focuses on structural chemistry.